# Eco-Friendly Tannin-Based Non-Isocyanate Polyurethane Resins for the Modification of Ramie (*Boehmeria nivea* L.) Fibers

**DOI:** 10.3390/polym15061492

**Published:** 2023-03-16

**Authors:** Manggar Arum Aristri, Rita Kartika Sari, Muhammad Adly Rahandi Lubis, Raden Permana Budi Laksana, Petar Antov, Apri Heri Iswanto, Efri Mardawati, Seng Hua Lee, Viktor Savov, Lubos Kristak, Antonios N. Papadopoulos

**Affiliations:** 1Department of Forest Products, Faculty of Forestry and Environment, IPB University, Bogor 16680, Indonesia; 2Research Center for Biomass and Bioproducts, National Research and Innovation Agency, Cibinong 16911, Indonesia; 3Research Collaboration Center for Biomass and Biorefinery between BRIN and Universitas Padjajaran, National Research and Innovation Agency, Cibinong 16911, Indonesia; 4Faculty of Forest Industry, University of Forestry, 1797 Sofia, Bulgaria; 5Department of Forest Products Technology, Faculty of Forestry, Universitas Sumatera Utara, Kwala Bekala, Medan 20145, Indonesia; 6Department of Agro-Industrial Technology, Universitas Padjadjaran, Jatinangor 40600, Indonesia; 7Department of Wood Industry, Faculty of Applied Sciences, Universiti Teknologi MARA (UiTM), Cawangan Pahang Kampus Jengka, Bandar Tun Razak 26400, Malaysia; 8Faculty of Wood Sciences and Technology, Technical University in Zvolen, 96001 Zvolen, Slovakia; 9Laboratory of Wood Chemistry and Technology, Department of Forestry and Natural Environment, International Hellenic University, 661 00 Drama, Greece

**Keywords:** *Acacia mangium* bark, cohesion strength, dimethyl carbonate, hexamethylenediamine, non-isocyanate, ramie fiber, tannin-based polyurethane

## Abstract

This study aimed to develop tannin-based non-isocyanate polyurethane (tannin-Bio-NIPU) and tannin-based polyurethane (tannin-Bio-PU) resins for the impregnation of ramie fibers (*Boehmeria nivea* L.) and investigate their mechanical and thermal properties. The reaction between the tannin extract, dimethyl carbonate, and hexamethylene diamine produced the tannin-Bio-NIPU resin, while the tannin-Bio-PU was made with polymeric diphenylmethane diisocyanate (pMDI). Two types of ramie fiber were used: natural ramie without pre-treatment (RN) and with pre-treatment (RH). They were impregnated in a vacuum chamber with tannin-based Bio-PU resins for 60 min at 25 °C under 50 kPa. The yield of the tannin extract produced was 26.43 ± 1.36%. Fourier-transform infrared (FTIR) spectroscopy showed that both resin types produced urethane (-NCO) groups. The viscosity and cohesion strength of tannin-Bio-NIPU (20.35 mPa·s and 5.08 Pa) were lower than those of tannin-Bio-PU (42.70 mPa·s and 10.67 Pa). The RN fiber type (18.9% residue) was more thermally stable than RH (7.3% residue). The impregnation process with both resins could improve the ramie fibers’ thermal stability and mechanical strength. The highest thermal stability was found in RN impregnated with the tannin-Bio-PU resin (30.5% residue). The highest tensile strength was determined in the tannin-Bio-NIPU RN of 451.3 MPa. The tannin-Bio-PU resin gave the highest MOE for both fiber types (RN of 13.5 GPa and RH of 11.7 GPa) compared to the tannin-Bio-NIPU resin.

## 1. Introduction

PU is a commercially synthesized polymer through polycondensation and exothermic reactions between molecules containing two or more isocyanate groups (-NCO) and polyol molecules containing two or more hydroxyl groups (-OH) [1]. The characteristics of PU (properties and structure) are determined by the structure of the molecule formed, which is influenced by the molecular weight, intermolecular forces, chain stiffness, cross-linking, and crystallinity [2]. The reaction between diisocyanates and diols can produce linear polymers, while using polyols with three or more hydroxyl groups will form cross-linked polymers. The polyols and isocyanates used in PU synthesis generally come from non-renewable raw materials, namely fossil-derived constituents [1]. The high demand for PU production has resulted in fossil resource depletion and has caused significant environmental problems, so more environmentally friendly renewable resources are needed [1]. Several studies have developed bio-based polyols derived from renewable and environmentally friendly raw materials. The sources of polyols currently being developed are compounds derived from triglycerides and sugars such as glucose, sucrose, sorbitol, and glycerol, which can be initiators in the synthesis of polyether polyols. Complete or partial replacement of polymer precursors derived from fossil raw materials (isocyanates, polyols, and other additives) with bio-based renewable raw materials is a feasible approach to increase the sustainability of polyol and PU production. Bio-polyol-based raw materials used in PU synthesis can be derived from natural phenolic compounds, carbohydrates, and vegetable oils [3]. In addition, tannins also represent alternative raw materials for bio-based polyols. The content of several functional groups in tannins, such as hydroxyl, allows tannins to form bonds with other compounds [4]. The potential application of tannins extracted from *Acacia mangium* bark for the production of tannin-based non-isocyanate polyurethane resin (tannin-based Bio-NIPU) needs to be investigated because mangium is an abundant, fast-growing, and multipurpose tree species that contains high-quality sources of tannins, so it will not cause competition with other food crops [5].

Tannins are natural polyphenolic components abundant in plants due to metabolic growth processes. Tannins have been widely used in various industrial applications, have higher reactivity than phenols, and can be used as an alternative raw material in bio-PU synthesis [6,7]. Tannins can be extracted from various plants, such as *Acacia mangium*. Tannins are extractive substances usually found in various plant parts, such as fruits, wood, leaves, and bark. The highest concentration of tannins is found in tree bark [7]. The extent of mangium plantation forests in Indonesia, which account for approximately 67% of the total area of mangium plantation forests worldwide, is one of the backgrounds of this research because the potential of the tree bark for the production of high value-added products still has not been fully explored due to some drawbacks, such as black coloration. Many industries need to make better use of tree bark, as it will only become waste. Therefore, in this study, pulp and paper industry waste in the form of mangium bark was utilized as a source of tannin extracts in the manufacture of tannin-based Bio-NIPU resins.

On the other hand, the toxicity of isocyanates is also a major challenge associated with serious human health effects, such as skin, eye, and throat irritation, difficulties breathing, and cancer. PU non-isocyanate resin synthesis can be performed by using cyclo-carbonate as an alternative material to replace toxic isocyanate. Two main methods are used to obtain cyclo-carbonate: the fixation of CO_2_ from the epoxy group and the reaction between the polyol and dimethyl carbonate. The epoxy ring can be produced by reactions between polyol hydroxyl groups or double-bond epoxidation reactions so that any compounds containing both functional groups can be utilized, such as tannins [8,9,10] and lignin [11,12], vegetable oils [13], etc. Das et al. (2020) carried out the synthesis of non-isocyanate PU based on carbonated soybean oil (CSO) and chitosan with the initial method of epoxidizing soybean oil and then carbonatation via the CO_2_ fixation method, which then produced CSO, which was then reacted with chitosan as polyamines via a cross-linking mechanism [13]. Isopropanolamine can also synthesize polyols with high hydroxyl values, which are dried and reacted with hexamethylene diisocyanate to produce bio-PU [14]. However, several studies related to the synthesis of non-isocyanate polyurethane still need improvement; for example, they do not use environmentally friendly raw material or produce toxic by-products that need further purification so that they are less economical. Thus, an alternative is needed in the synthesis of tannin-based non-isocyanate polyurethane resins. Following the findings reported by Sunija et al. (2014) and Thébault et al. (2017), the synthesis of bio-PU can be based on the reaction of the hydroxyl groups of tannin flavonoid units with dimethyl carbonate (DMC), which can then be reacted again with hexamethylenediamine (HMDA), resulting in a PU structure [9,15]. Therefore, this study synthesized tannin-based Bio-NIPU resin by reacting tannins using DMC and HMDA, which are more environmentally friendly, non-toxic, and do not produce toxic by-products either.

On the other hand, Indonesia has high potential for producing ramie (*Boehmeria nivea* (L.) Gaudich) fiber due to the high demand for fiber as an alternative to cotton, superior varieties of seeds, and ample land for growing ramie. Ramie fiber has good strength and stiffness properties, decay resistance, and has twice the strength of ordinary wood fiber [16]. Although ramie fiber has several advantages and is more environmentally friendly than synthetic fibers, there are also some disadvantages, such as the relatively high water absorption, which can lead to weak interfacial bonds between the polymer matrix and fibers, resulting in decreased mechanical properties. In addition, ramie fiber also has relatively low fire retardancy, so additives such as halogen, nitrogen, and phosphorus-based compounds are needed to increase its heat resistance in polymer composites [17]. Therefore, this research aimed to modify ramie fiber to increase its resistance to heat by impregnation using tannin-based Bio-NIPU and compare it with a tannin-based Bio-PU resin, which still contains isocyanate.

## 2. Materials and Methods

Mangium bark used in this study (*A. mangium*, 40–60 mesh) was obtained from Tanjung Enim Lestari Pulp and Paper Company (Palembang, Indonesia). Aquades, hydrogen peroxide (7%, Merck, Darmstadt, Germany), isopropyl alcohol 4%, urea 1%, pMDI (±31 NCO content, 99.6% solids content, BASF), dimethyl carbonate (DMC 99%, Sigma-Aldrich, Hamburg, Germany), and hexamethylenediamine (HMDA, Sigma-Aldrich, Hamburg, Germany) were bought from a local supplier for the preparation of tannin-based Bio-PU and Bio-NIPU resins. Degummed ramie fibers were purchased from CV. Rabersa (Wonosobo, Indonesia).

### 2.1. Tannin Extraction

Mangium bark was obtained from factory waste. Mangium bark was first pollinated using a dish mill and hammer mill with a sieve measuring 40 mesh and retaining 60 mesh. Tannins were produced through an extraction process by the maceration method with a sample:solvent (aquades) ratio of 1:10. Then, 100 g of mangium bark powder was placed in a 2 L Erlenmeyer flask, and 1 L of distilled water was added and they were heated at 60 ± 2 °C for 6 h. Furthermore, the extract was squeezed, and the filtrate was collected; the resulting residue was then redissolved using distilled water with the same ratio of 1:10, stirred, and reheated at the same temperature and time. This step was repeated until the extracted filtrate was clear or processed four times. The resulting filtrate was then stored in a showcase with a temperature ranging from 2 to 8 °C and concentrated using a rotary evaporator with a vacuum pressure of 74 mbar, a dynamic set of 58 mbar, a temperature of 60 ± 2 °C, and a speed of 100 rpm to produce a viscous tannin extract.

### 2.2. Determination of Tannin Solid Content

The petri dish was placed in the oven at 105 ± 2 °C for 4 h. After this, it was placed in a desiccator for 30 min. The weight of the empty petri dish was determined and recorded. Then, 1 g of the viscous tannin extract sample was weighed in a petri dish and placed in the oven for 3 h at 105 ± 2 °C. The sample was placed into the desiccator for 30 min and weighed again after the oven.

### 2.3. Preparation of Tannin-Based Bio-PU and Tannin-Based Bio-NIPU Resins

Tannin-based Bio-PU (tannin-based Bio-PU) was prepared by reacting tannin powder and pMDI. The tannins extracted from hot water were dissolved in a 20% NaOH solution with a ratio of 1:10 (*w*/*v*), and then 80% pMDI, which had been dissolved in acetone with a NCO/OH mole ratio of 0.3, was added by dripping little by little, and then polymerized for 5 min. The mixture was stirred mechanically with a stirring speed of 500 rpm. The final polymerization was carried out at room temperature for 30 min, and the viscosity was measured at room temperature with a rotational rheometer.

Tannin-based Bio-NIPU (Tannin-based Bio-NIPU) was prepared by the reaction between tannin and DMC with various ratios of tannin:DMC (1:1 and 1:2) at 50 ± 2 °C and stirring for 15 min. HDMA was added to the sample little by little with a sample ratio of tannin:HMDA (4:1) with a correction factor for the results of the solid content of the tannin-DMC. The mixture was stirred mechanically with a stirring speed of 500 rpm for 30 min, and its viscosity was measured at room temperature with a rotary rheometer.

### 2.4. Preparation of Ramie Fiber

Two types of ramie fiber were used, natural ramie fiber and pre-treated ramie fiber. Natural ramie fiber (RN) was degummed ramie fiber, while pre-treated ramie fibers (RH) were prepared by soaking in H_2_O_2_ and isopropyl alcohol. The first stage was soaking in a mixture of 7% H_2_O_2_ and 1% urea at 85 °C for 60 min. The second stage was soaking at the same temperature and duration in 4% isopropyl alcohol.

### 2.5. Impregnation of Ramie Fiber with Tannin-Based Bio-PU and Tannin-Based Bio-NIPU Resins

Natural ramie fiber (RN) and H_2_O_2_ (RH_2_O_2_) treated ramie fiber were impregnated with tannin-based Bio-PU and tannin-based Bio-NIPU resins, respectively. Impregnation was conducted in a 1 L vacuum chamber with a 2-stage vacuum pump (VC0918SS, VacuumChambers.ue., Białystok, Poland). The initial weight of flax fiber was recorded before vacuum impregnation. Each sample of 3 g of ramie fiber was soaked in 30 mL of tannin-based Bio-PU and tannin-based Bio-NIPU resins impregnated at 27 ± 2 °C under 50 kPa pressure for 1 h. The impregnated fiber was then dried at room temperature 25 ± 2 °C for 24 h. Dry ramie fibers were weighed to determine the weight gain after impregnation. Weight gain (%) was calculated by dividing the difference in ramie fiber mass after and before impregnation by the initial ramie fiber mass. The impregnated ramie fibers were then stored in zip-lock plastic bags for further testing [18].

### 2.6. Characterization of Viscous Tannin Extracts, Tannin-Based Bio-PU, and Tannin-Based Bio-NIPU Resins

FTIR (Spectrum Two, Perkin Elmer, MA, USA) with the Universal Attenuated Total Reflectance (UATR) method was used to analyze the functional groups of each sample. The average accumulation was recorded in as many as 16 scans at a resolution of 4 cm^–1^ with wavenumbers ranging from 4000 to 400 cm^–1^. Measurements were made at a temperature of 25 ± 2 °C. Thermogravimetric analysis was performed using a TGA instrument (TGA 4000, Perkin Elmer, MA, USA) to determine the heat stability of the samples. First, 20 mg of each sample was weighed into a standard ceramic crucible and heated in a nitrogen atmosphere at a 20 mL/min flow rate. The temperature used in heating ranged from 25 to 750 °C with a heating rate of 10/min. Percent weight loss, weight loss rate, and residue were analyzed with the help of Pyris 11 software (Version 11.1.1.0492, Pyris, MA, USA). A rotational rheometer was used for viscosity analysis. Here, 20 mL of each sample was placed into a special measuring cup (C-CC27, Anton Paar, Buchs, Austria) and positioned on a rotational rheometer (RheolabQC, Anton Paar, Buchs, Austria). Viscosity measurements were carried out with spindle CC no. 27 with a constant shear rate of 250/s at 25 °C for 120 s to determine the average viscosity. Viscosity results were displayed with the RheoCompass app (Version 1.33, Anton Paar, Buchs, Austria).

### 2.7. Examination of Ramie Fiber Properties

The diameter of each ramie fiber was measured using ten fibers. The measurement was conducted with a light microscope (BX63, Olympus, Tokyo, Japan) and imaging software (Labspec 6, Horiba, Kyoto, Japan). The weight gain of ramie fibers was calculated by dividing the post-impregnation mass of ramie fibers by the initial mass of ramie fibers. Using FTIR spectroscopy (SpectrumTwo, Perkin Elmer Inc., Hopkinton, MA, USA) coupled with the UATR method, similar to viscous tannin extracts, tannin-based Bio-PU, and tannin-based Bio-NIPU resins, was used to investigate the changes in the functional groups of the original and impregnated ramie fibers. Thermogravimetric analysis of RN and RH fibers before and after impregnation was performed using a TGA instrument (TGA 4000, Perkin Elmer, MA, USA). Here, 20 mg of ramie fiber was weighed into a standard ceramic crucible and heated in a nitrogen atmosphere at a 20 mL/min flow rate. The temperature used in heating ranged from 25 to 750 °C with a heating rate of 10/min. Percent weight loss, weight loss rate, and residue were analyzed with the help of Pyris 11 software (Version 11.1.1.0492, Pyris, MA, USA). Tests for the tensile strength of RN and RH fibers before and after impregnation were carried out based on the ASTM D 3379–75 standards using UTM [19]. The specimens used were single fibers separated from strand bonds. The specimen length was between 20 and 30 mm, and the overall fiber length was approximately three times the length of the specimen. The specimens were tested with a loading of 5000 N and a crosshead speed of 1 mm/min. The test was conducted in a room with a temperature of 25 ± 2 °C.

## 3. Results and Discussion

### 3.1. Characteristics of Mangium Powder and Tannin Viscous Extract

The characteristics of mangium powder and tannin extract include the water content of mangium powder, solid content, and yield (Table 1). The moisture content of mangium bark powder used for tannin extraction was determined by measuring the weight of free water that was evaporated and was not tightly bound in the material network using heat. The water content of the resulting mangium bark powder was 5.47 ± 0.13% (Table 1); this water content was lower than the water content reported in the study by Mutiar et al. (2019) of 16.76 ± 1.61% and the water content in Liu et al.’s research (2020) of 10.1% [20,21]. The water content can reveal the resistance and the best storage technique of the material to minimize the influence of microorganism activity. The low water content in the material makes the material more resistant to the activity of microorganisms or fungi during storage, because water can be a medium for microbial growth and other chemical reactions [22,23]. The mangium bark powder was used as a tannin extraction material using the maceration method with hot water. The maceration method was chosen because it is simpler and cheaper but requires many solvents. Maceration is more effective because tannins are extractive substances limited to settling in the cell cavities without forming chemical bonds with other wood cell wall constituent components, such as lignin, cellulose, and hemicellulose. Therefore, extractive substances are more accessible than other components and do not require more complicated extraction methods. In addition, the solvent selection used in the extraction process here uses water compared to organic solvents due to cost considerations and environmental impacts. Using organic solvents in large quantities is not only uneconomical but can also be harmful to the environment if the waste is not managed properly, so the use of water with a temperature of 60–70 °C was selected as the optimal choice for extraction in this study. The hot water used will penetrate the cell wall and enter the cell cavity, which contains extractive substances (tannins), so that the tannins will dissolve due to the difference in the concentration of the active substance solution inside the cell with the solution outside the cell, and the solution will be pushed out. There are two stages in the extraction process: the first stage is an osmosis process performed in a short time, and the second stage is an osmosis and diffusion process that coincides [24]. Cuong et al. (2020) stated that the use of high temperatures in the extraction process could optimize the yield obtained, due to the possibility of decomposition of proteins and cell structures during the heating process, so it is hoped that the bonds of tannins with proteins can be optimally broken and increase the yields of tannin extracts [24].

The yield percentage of tannin viscous extract in this study was 26.43 ± 1.36%; the results of the study on the tannin extract yield showed that the amount of hot water-soluble extractive from mangium bark used was greater than that of Sosuke et al. (2003) of 18% with the same ratio of powder:solvent used, namely 1:10 [25]. This difference could be due to the age of the wood used, so the older the plant, the higher the extractive content [26]. However, this study could not prove this directly due to the absence of plant age data. In addition, the content of extractive substances can also be affected by the location and environmental conditions in which the plants grow, so the yields obtained vary (Table 1). The extractive yield in this study was a brownish viscous filtrate, which was then determined for its solid content, which was 93.46 ± 0.41% (Table 1). The solid content obtained was higher when compared to the results of the study by Liu et al. (2020), which was 89.8% [21]. Solid content is an essential parameter in resin synthesis, which can affect the drying time, the effect of moisture on the impregnation medium, and the weight gain of the media used after being impregnated with resin. The high content of solids or tannins (<50%) may indicate that tannins have potential as a modification material for polyurethane resins, replacing commercial polyol structures to form urethane bonds. The tannin solid content can affect the distribution of the resin and the formation of a resin layer during the impregnation process, so it has a vital role in the application of the impregnation process. The low solid content of tannins can result in the resin not being successfully synthesized because the water content is too high, so too much moisture must be removed during the resin synthesis. In addition, it can also affect the absorption and adhesion of the resulting resin when it is used as an impregnant; the resin will quickly detach or leach so that the bond formed is unstable between the impregnant and the impregnation medium. The high solid content of tannin as a resin synthesis material can cause an increase in the viscosity value of the resin so that when the resin is used as an impregnant, it can increase the percentage of weight gain in the impregnation medium used [27]. This is caused by the large chemical structure of polyphenol tannins, so the viscosity and solid content of the resulting resin also increase [28].

Confirmation of the tannin in solid and viscous liquid using FTIR produced the spectrum shown in Figure 1a. The characteristic spectrum of tannins from mangium showed the presence of several absorption bands at specific wavenumbers. Indirectly, the two spectra have almost the same absorption bands, but the absorption peaks in solid tannins are sharper and more visible than in the viscous liquid form. The absorption peak at wavenumber 3325–3300 cm^−1^ indicated absorption from the vibrational strain of the -OH group [29], and the broad peak indicated the presence of hydrogen bonds formed [15]. The presence of the -OH absorption peak was due to the water content contained in the viscous tannin extract because the solid content contained was 93.0%, so it can be ascertained that there was a water content of 7.0%. Then, at wavenumber 2840 cm^−1^, it showed a C-H stretching vibration from the -CH_2_ and -CH_3_ bonds in the ester group [18]. In addition, the absorption peak at 2135 cm^−1^ indicated stretching of the C-H bonds of the aromatic components originating from the tannin structure. It is also reinforced by an absorption band at wavenumber 1600 cm^−1^, which is the vibration of the tannin structure’s aromatic ring strain [30]. This is in agreement with Thebault’s research (2017), which stated that the stretching vibration of the C-H bond from the methyl, methyl, and aromatic methylene groups, as well as the stretching of the C-O bond from the ester group at wavenumbers 1630–1600 cm^−1^, led to the formation of a strong resonance on the tannin aromatic ring [9]. The spectrum of the tannin extract also showed absorption bands at wavenumbers 1455 cm^−1^ and 1034 cm^−1^, respectively, indicating the presence of a methyl bond (C-H strain from the methylene group) and C-O-C ester bonds in the heterocyclic ring of the flavonoid group [18].

The thermal stability of tannins was determined based on the TGA/DTG thermogram analysis shown in Figure 1b. The TGA/DTG thermogram analysis of tannins was determined under a nitrogen atmosphere and can show the relationship between the mass change or tannin degradation and a given temperature increase. The tannin decomposition process occurred in two stages; the first stage was the process of water evaporation and the second, more significant stage, was the tannin degradation stage. Tannins lost weight by 10% at 109.9 °C, which can be attributed to the loss of absorbed moisture or the occurrence of water evaporation processes but does not involve the degradation of the tannin structure [31]. Based on these results, the tannins produced in this study were more stable when compared to tannins from cashew seeds, which experienced a weight loss of ±10% at a lower temperature range, namely 29.5–60.0 °C [32]. Around 25% weight loss occurred at 311.6 °C, while, at 562.2 °C, a 50% weight loss occurred due to the decomposition of the aromatic rings and it had a decomposition peak at 744.4 °C with a residue level of 37.6%. The stability of the tannins produced in this study was better when compared to tannins from pomegranate barks, which began to decompose at 149.0 °C with a residue level of 36.4%. This could be due to the more significant amount of carbohydrate content in pomegranate bark extract, mainly due to the presence of sugar in the resulting tannin structure, so that the tannin from mangium bark is more thermally stable with a residue level of 37.6% [30]. This shows that the thermal stability properties of tannin extracts vary depending on the plant source and the extraction process used. The existence of an aromatic structure in a complex tannin extract causes high heat resistance, so tannins can be used as an alternative to polyols in synthesizing polyurethane resins with high-temperature applications due to the presence of complex aromatic structures and cross-links formed during the synthesis process.

### 3.2. Characteristic of Tannin Bio-PU and Tannin Bio-NIPU Resins

Figure 2a shows that the tannin extracts, pMDI, and tannin-based Bio-PU samples had peak intensity at wavenumbers 3800–300 cm^–1^, a hydroxyl group strain vibration [29]. The broad peak shown indicated the presence of hydrogen bonds that may occur due to the existence of polyphenols and polysaccharides in the tannin extract, which can also come from NaOH, which is used as a solvent in the synthesis of tannin-based Bio-PU [15,18]. The results showed two absorption bands at wavenumbers 2920 cm^–1^ and 2840 cm^–1^, which were C-H stretching vibrations of the -CH_2_ and -CH_3_ bonds of the esterified product in the three types of samples, although there was a shift in the tannin extract samples [29]. Furthermore, in the tannin extract, there was also an absorption peak at wavenumber 1600 cm^–1^, indicating a vibrational strain from the aromatic ring of the tannin structure [30]. The pMDI spectrum resulted in a strong absorption peak at 2240 cm^–1^, characteristic of the N=C=O stretching of the isocyanate groups. The absorption bands 1765 cm^–1^ and 1710 cm^–1^, respectively, showed strong C=O elongation of carboxylic acids and C=O stretching of aliphatic ketones. In addition, vibrational strains of the N-O and O-H bending groups of the carboxylate groups occurred at wavenumbers 1515 and 1425 cm^–1^, respectively. Tannin-based Bio-PU showed a shift in the absorption band at wavenumber 1615 cm^–1^ (urethane bond), which indicated that the resin was successfully synthesized due to the C=O bond of the polyurethane amide, as confirmed by the presence of an absorption band appearing in the fingerprint area of 810 cm^−1^ according to C-H bending [21].

Figure 2b provides a typical spectrum of the tannin-based Bio-NIPU synthesis process. The synthesis of tannin-based Bio-NIPU uses dimethyl carbonate and hexamethylenediamine to form the characteristic polyurethane group, NCO-OH (urethane). The mechanism for the formation of tannin-based Bio-NIPU consisted of two stages; the first one was by reacting a viscous tannin extract with DMC so that carbonated tannin could be obtained, where the spectrum can be seen in the tannin-DMC section. Several new absorption peaks appear when tannins are carbonated, both on the viscous tannin extract and tannin-DMC, which had absorption peaks at wavenumbers 3360–3325 cm^–1^ (OH strain vibration). Nonetheless, there was a shift in tannin-DMC due to the reaction between the viscous tannin extract and dimethyl carbonate. There was a stretching vibration of the C-H bond from the methyl group, methylene aromatic, and stretching of the C-O ester at wavenumbers 1637–1631 cm^–1^, resulting in a resonance process in the aromatic ring [9]. Alkane functional groups in the structure of viscous tannin extract, tannin-DMC, and tannin-based Bio-NIPU can be confirmed by the absorption peak at wavenumber 1455 cm^–1^ (C-H bending vibration of the methylene group). The characterizing group for the type of viscous tannin extract was found in the absorption band 1338 cm^–1^ (OH bending vibration of the phenolic group) and 1162 cm^–1^ (straining vibration C-O ester), which can prove that the tannin used is a derivative of gallic acid (a monomer of hydrolyzed tannin). A new absorption band at wavenumber 1752 cm^–1^ (carbonyl group) in tannin-DMC indicated a tannin carbonation reaction. The intensity of the hydroxyl groups in tannin-DMC also decreased due to the response of any hydroxyl groups in tannin with DMC [33,34]. Absorption bands of 1637 cm^–1^ (strain vibration of the aromatic ring) and 1275 cm^–1^ with high-intensity absorption peaks (straining vibrations of the C-O ester group) were reinforced by absorption bands found in the fingerprint region, indicating that the reaction between tannins and DMC causes tannins to be carbonated and undergo fragmentation. The second stage of the tannin-based Bio-NIPU synthesis was reacting with the electropositive carbonyl group to bind to the amine group and form a urethane group. In tannin-based Bio-PU, the group comes from isocyanate (pMDI) used in the synthesis process. There was a new absorption band at wavenumber 2853 cm^–1^ (methylene C-H stretching vibration of the ester product) [29]. The absorption peak is 1643 cm^–1^ (CO group and vibration from the amide group), indicating a polymerization reaction in forming urethane bonds [9]. The tertiary alcohol group of the resin is confirmed from wavenumber 1156 cm^–1^. The opening of the C-O-C ether bond occurred in the group heterocyclic ring tannin flavonoids shown in the absorption bands at 1018 and 1029 cm^–1^ due to the formation of new bonds due to the reaction process between tannins and DMC and HMDA, resulting in the stretching of the main hydroxyl groups in carbohydrates shown at wavenumber 1105 cm^–1^ [9].

Based on the two types of FTIR functional group analysis, it can be shown that the viscous tannin extract had a hydroxyl group, which can serve as an alternative to polyol to bind to the isocyanate group of the pMDI used to produce a urethane identifying group (tannin-based Bio-PU). However, it is still toxic and not environmentally friendly, so the isocyanate material based on the FTIR results above can be confirmed to be substituted using DMC material to produce carbonated tannins as a carbonyl source to bind with the amine group of HMDA to obtain the desired identifying group at the end of the synthesis, namely the urethane group (NCO). Figure 3 (tannin-based Bio-PU) and Figure 4 (tannin-based Bio-NIPU) show the possible reaction schemes for the two types of resin.

Because the resulting tannin-based Bio-PU and tannin-based Bio-NIPU resins will be applied as fiber modification materials through an impregnation process, viscosity data and cohesive strength are required. The obtained viscosity and cohesion strength data can be compared in Figure 5. Viscosity is an important characteristic because it relates to the viscosity of a liquid, which can affect fluid flow when applied as an impregnation material into fibers. The greater the viscosity value produced, the greater the thickness of a material, so it will be more difficult for the material to flow due to the slow movement of the sample liquid particles. At the same time, the cohesive strength is vital to see the internal strength of the resin due to interactions between materials that bond together [35]. The bond between the resin and the fiber used later will tend to be weaker if there is cohesion damage or internal damage to the resin itself; therefore, this test is essential to analyze. Figure 5 shows that each sample’s viscosity and cohesive strength have a positive correlation, where the viscous tannin extract has the lowest viscosity and cohesive strength values, followed by tannin-based Bio-NIPU and the highest tannin-based Bio-PU. The tannin viscosity was 1.60 mPa·s, while tannin-based Bio-NIPU (20.35 mPa·s) and tannin-based Bio-PU (42.70 mPa·s) (Figure 5a) differed. This was due to a more complex bonding interaction in tannin-based Bio-NIPU and tannin-based Bio-PU compared to tannins due to using other mixed materials when synthesizing the two resins. The higher viscosity value of the tannin-based Bio-PU resin compared to tannin-based Bio-NIPU was due to the use of pMDI in tannin-based Bio-PU, which had a reasonably high viscosity.

The high viscosity value of the resin will affect the impregnation process that is carried out later; if the resin viscosity value is too high, it can result in the resin only sticking to the surface of the fiber, not being absorbed and bonding with the fiber components ideally, because the viscosity value is proportional to the cohesion strength value. Cohesion strength is also determined from the chemical composition of the resulting resin due to its interaction and molecular strength. The cohesive strength of the two resins is higher when compared to the viscous tannin extract due to the formation of cross-links in both short-chain and long-chain molecular bonds and the formation of three-dimensional networks in the resin molecular chains that occur during the polymerization process. The cohesion strength of the viscous tannin extract (0.40 Pa), tannin-based Bio-NIPU (5.08 Pa), and tannin-based Bio-PU resin (10.67 Pa) is presented in Figure 5b. The viscosity and cohesive strength values of tannin-based Bio-PU tend to fluctuate due to using pMDI, which hardens quickly. Thus, the viscosity and cohesive strength decreased at the end of the testing compared to at the start. The reaction temperature can affect the value of cohesion strength; the higher the temperature used, the higher the cohesive strength.

The thermal stability of the two resins was evaluated based on the thermogram results in Figure 6. The initial weight loss of tannin-based Bio-PU occurred at a temperature of 96.5 °C, caused by the evaporation of water and chemicals in the tannins [36]. At 100 °C, the weight loss of tannin-based Bio-PU resin is 0.40 %/°C. The weight loss at around 280.0 °C reached approximately 0.20 %/°C, which indicated the decomposition process of the urethane bond contained in the resin. Further oxidative degradation of urethane occurs at around 450.0 °C with a minimal weight loss of only ~0.02%/°C [37]. The degradation of the polymer’s three-dimensional structure, the decomposition of aromatic tannin rings, and the main oxidative processes of tannins occurred at temperatures above 450.0 °C [37,38,39]. The residue of the tannin-based Bio-PU resin was 64.7%, and there was only a weight loss of around 35.3%; this was due to the use of pMDI, a type of thermosetting resin.

On the other hand, for the tannin-based Bio-NIPU resin, the initial weight loss occurred at a temperature of 201.1 °C; this temperature was higher than that of the tannin-based Bio-PU resin, which was due to the process of evaporation of water and the chemicals contained therein, which did not form bonds intensely during the polymerization process [40]. The most significant degradation occurred at a temperature range of 450.0 °C with a decrease in the weight of the resin reaching 1.20%/°C. This was due to the degradation process of the urethane bonds in the resin; temperatures above 450.0 °C only resulted in a relatively small weight loss, around 0.03%, indicating the presence of decomposition of the aromatic ring of tannins.

### 3.3. Characteristics of Ramie Fiber

The different types of ramie fiber used are ramie fiber without treatment and ramie fiber that has undergone a pre-treatment using H_2_O_2_, so that the fiber becomes softer and has a lighter color. However, based on the FTIR spectrum (Figure 7), it can be confirmed that there was no significant difference between the two types of fibers. The result illustrates that the softening process did not significantly change the fiber content. Some of the absorption bands that appeared are the vibrational strain of the hydroxyl group (3333 cm^–1^), and the absorption band at wavenumber 2898 cm^–1^ showed the vibrational strain of the methyl group from the chemical components of cellulose and hemicellulose contained in ramie fiber [18]. Then, in natural ramie fiber (RN), there was a peak of C-H vibration strain absorption from carbohydrates (1607 cm^–1^), while, in pre-treated ramie (RH), it experienced a shift due to reactions with other chemicals when the fiber softening process was carried out, which was at the wavenumber of 1635 cm^–1^. Then, both RN and RH fibers showed absorption peaks, including vibrational strains from the carbonyl and hydroxyl groups of polysaccharides or cellulose fibers (1028 cm^–1^).

The impregnation technique was carried out to incorporate resin material into ramie fiber, aiming to improve the ramie’s thermal and mechanical properties and compare the ramie before and after impregnation. FTIR analysis was performed to confirm the functional groups of ramie before and after impregnation with each type of resin and different ramies for 1 h. Figure 8 shows the functional group analysis of natural ramie (RN) impregnated with two different resin types. Natural ramie (RN) impregnated with tannin-based Bio-PU (RN tannin-based Bio-PU) and tannin-based Bio-NIPU (RN tannin-based Bio-NIPU) had a new absorption peak at wavenumber 2940 cm^–1^ that indicated the presence of N-H groups, because the resin used for impregnation penetrated well into the RN ramie. The carbonyl group was also confirmed at the absorption band of 1700 cm^–1^ in tannin-based Bio-PU RN, and a shift occurred at 1688 cm^−1^, indicating a bond between the RN ramie and the resin. Wavenumber 1515 cm^–1^ indicated the presence of primary and secondary C-N amide groups originating from the resin structure, while the vibration of the C-O group occurs at wavenumber 1260 cm^–1^. The C-O-C ether bond structure strain was confirmed from the peak absorption at wavenumber 1030 cm^–1^. The NCO group was established in the range 820–780 cm^–1^, indicating that the resin had successfully permeated and bonded with the hydroxyl groups in the ramie to form a urethane bond.

Figure 9 shows the FTIR spectrum of ramie pre-treated (RH) before finally being impregnated with tannin-based Bio-PU and tannin-based Bio-NIPU resins. There were absorption peaks that appeared as in RN tannin-based Bio-PU and RN tannin-based Bio-NIPU, such as at 2942 cm^–1^, which indicated the stretching of the amine groups of the resin impregnated into the ramie. Stretching the methyl groups of carbohydrates (1689 cm^–1^), the activity of the C-N groups was derived from the amide in the resin used (1528–1313 cm^–1^). The wavenumbers 1260–1104 cm^–1^ had absorption peaks from the C-O vibrations of tannin-based Bio-PU and tannin-based Bio-NIPU. There was also stretching of the C-O-C ether bond at wavenumber 1052 cm^–1^. Figure 8 and Figure 9 show that pre-treatment of ramie did not change its ability to absorb resin during the impregnation process; even the RH tannin-based Bio-NIPU tended to give a higher intensity of absorption peaks when compared to RN tannin-based Bio-NIPU. In addition, it can also be confirmed that based on the functional group analysis carried out, the absorption peaks between the two types of resin were almost the same. There was no significant difference because there was still a typical absorption peak as a marker for the presence of urethane groups in the two types of resin applied to the ramie. Therefore, it can be concluded that tannin-based Bio-NIPU resin can be an alternative to isocyanate (pMDI) in commercially used synthetic polyurethane resins. The reaction mechanism between the ramie that binds the resin during the impregnation process can be seen in Figure 10.

The stability of the two types of ramie fiber before impregnation was determined based on the analysis of the TGA-DTG results shown in Figure 11. The weight loss in ramie occurs due to the heat given as the temperature increases. The initial process of water loss in the fiber and the decomposition of components with low molecular weights started at 25.0–100.0 °C, so the reduction was relatively small. Weight loss of 10% occurred at RN (289.3 °C) and RH (297.8 °C); then, at temperatures above 300.0 °C, there was a significant decrease so that the decrease in the TGA thermogram appeared extreme (Figure 11). Fiber decreased by 25–50% in RN, which occurred at a temperature of 345.0–367.5 °C, and RH at a temperature of 343.7–359.2 °C, due to the degradation process of chemical components in the ramie fiber, in the form of lignin, cellulose, and hemicellulose. Lignin begins to degrade at 280.0–500.0 °C, cellulose glycosidic bonds decompose at 300.0–400.0 °C, and hemicellulose, the chemical component most easily degraded by heat, degrades at 200.0–290.0 °C [41,42,43]. At temperatures of 360.0–750.0 °C, it was confirmed that there was a decomposition process of lignin structures with high molecular weights [39,43]. At the end of the heating process, RN had a residue of 18.9%, while RH was 7.3%. This shows that RH experienced a more significant reduction in weight up to 92.7%, so, from the TGA and DTG results obtained from the two types of fiber, it can be concluded that RN fiber is more resistant and stable to heating than pre-treated ramie fiber (RH).

The thermal stability of RN and RH fibers impregnated with the two types of resins also decreased in weight (Figure 12 and Figure 13). Two reaction processes occurred based on the formed thermogram peaks. The initial stage at a temperature of 60–110 °C was associated with evaporating the water contained in the fiber, with an average decrease of 2.0%/°C. Extreme peaks marked the following process due to the degradation of large molecules, namely the chemical content in the fiber, which is a lignocellulosic group at temperatures above 300.0 °C. In RN fiber after impregnation, there was a weight loss of 50.0% starting from a temperature of 313.1–359.5 °C, while, in RH, it occurred after impregnation at 329.8–359.7 °C. However, when viewed from the perspective of the temperature, RH underwent a degradation process at a higher temperature than RN. Meanwhile, when viewed from the result of total weight loss at the end of the heating process, RN generally has a smaller weight loss when compared to RH. RN impregnated with tannin-based Bio-PU resin experienced a total weight loss of 69.4% (30.5% residue), while RN-Bio-NIPU had a total weight loss of 80.9% with a residue of 19.1% (Figure 12).

On the other hand, Figure 13 shows that RH impregnated with tannin-based Bio-PU had a higher residue at the end of the heating process than RH-Bio-NIPU, which had residues of 24.3% and 16.2%, respectively. Based on these results, it can be concluded that the determination of thermal stability can be obtained from the amount of weight loss experienced by each sample and the residue left behind at the end of the heating process. The thermogram results in Figure 12 and Figure 13 show that RN fiber exhibited better thermal stability due to the use of H_2_O_2_ and isopropyl alcohol (in the pre-treatment of ramie fiber), which caused a strong oxidation and degradation effect on the cellulose chain because the two materials used can degrade large molecules into smaller ones, as well as delignification and gum removal processes. Therefore, the degradation of the fiber constituent components makes RH’s thermal stability smaller than that of RN without pre-treatment [44]. In addition, when compared between the two types of resin, tannin-based Bio-PU resin was characterized by better thermal stability than tannin-based Bio-NIPU. This was attributed to the use of isocyanate (pMDI) in the synthesis process because isocyanate offers good thermosetting. Hence, its thermal stability is still higher than that of non-isocyanate resins. However, the results obtained are only 8.0–10.0% different, so, taking into account the side effects of using isocyanates in sizeable industrial production, the use of tannin-based NIPU is still relatively efficient and can be subjected to further development so that the results of thermal stability are the same or better than for resins that still contain isocyanates. Based on the analysis of its thermal properties, it can be seen that the impregnation process can increase the thermal stability of the fiber due to the excellent process of incorporating the thermosetting resin into the fiber, resulting in the formation of a bond between the resin and the fiber, which increases the thermal stability of the impregnated fiber compared to the non-impregnated fiber.

In addition to confirming the success of the impregnation process through FTIR analysis, it can also be achieved by calculating the resulting weight gain after the ramie is impregnated. If the resin has penetrated and bonded well with the chemical components in the ramie, it is characterized by a high weight gain, and vice versa: a low weight gain indicates that the impregnation process is not optimal. Table 2 shows the weight gain for each type of ramie impregnated with the two different resin types. When compared to the type of resin used, it can be seen that the tannin-based Bio-NIPU resin produced a higher weight gain when compared to the tannin-based Bio-PU resin when impregnated with RN (21.2%) and RH (26.5%). This follows the results of the viscosity and cohesive strength, where the viscosity and cohesion strength values of tannin-based Bio-NIPU are lower than those of tannin-based Bio-PU. This makes the resin fluid flow easier when impregnated into ramie fibers, and causes the resin to be adsorbed optimally because of its lower viscosity.

In contrast, the tannin-based Bio-PU resin, with a higher viscosity and cohesive strength value, gives high viscosity to resin, which causes the resin flow to be difficult to absorb into the ramie, and the bond between the resin and the chemical components in the ramie is not optimal. This resulted in the resin only coating the ramie surface so that the resin would easily leach. In addition, the weight gain results were also indirectly confirmed from the FTIR spectra results, where the absorption peak intensity produced by the tannin-based Bio-NIPU resin was higher than for tannin-based Bio-PU. Besides the viscosity and cohesion strength, the weight gain results are influenced by the cavity or ramie diameter and the impregnation time. Furthermore, when compared based on the type of ramie used, pre-treated RH had a higher average weight gain than RN when impregnated in tannin-based Bio-PU and tannin-based Bio-NIPU resins (Table 2). The use of isopropyl alcohol and hydrogen peroxide can reduce the latex content of the ramie fibers, as well as other non-cellulose degradation due to oxidation, and results in much softer fibers [44]. Eventually, it is easier to bind and absorb resin materials when impregnation is carried out for pre-treated ramie fibers (RH).

The application of impregnated ramie as a functional textile material causes the need to test its mechanical properties by analyzing its tensile strength and modulus of elasticity. Analysis of the mechanical properties of the ramie fiber can be seen in the graph provided in Figure 14 and Table 3. Based on the graph, the tensile strength of the modified ramie through the impregnation process had increased compared to the unmodified fiber. Table 3 shows that the highest ramie tensile strength results were obtained from ramie impregnated using tannin-based Bio-NIPU at RN (451.3 MPa) and RH (426.1 MPa). The data of the test results showed that the impregnation process and the type of resin and fiber used had significantly affected the tensile strength of ramie fiber before impregnation. However, no interaction was found between the two types of factors. The tensile strength results indicate the ability of the ramie to withstand the longitudinal stress imposed on the fiber [42].

The increase in the tensile strength value was caused by the abundance of tannin molecules synthesizing the resin, causing many tannin phenolic OH groups to bond to form urethane groups. This caused the chain linkages and cross-links in the resin and ramie, increasing their mechanical strength. The tannin-based Bio-PU resin had lower tensile strength due to the bulky structure of the pMDI used in making the resin. In addition, the high reactivity of pMDI causes the resin to harden more efficiently, even at room temperature, so the bond between tannin-based Bio-PU and the ramie is not optimal; the resin only sticks to the surface of the ramie. It causes the ramie to become stiff and easily broken, resulting in low tensile strength. This is in line with the research of Lee et al. (2020), which stated that the mechanical strength of a material is influenced by its flexibility, the strength of chain bonds, cross-links, and the number of rigid moieties [45].

Moreover, tensile strength is related not only to the chemical composition of the ramie but also to its internal structure and morphological characteristics. Ramie has great cellulose content and a relatively high microfibril angle, resulting in superior tensile strength to other cellulose ramies [42]. The tensile strength results of the RN fiber type were higher compared to RH. This makes it possible for the oxidation and degradation reactions to be too strong due to the use of peroxide in the pre-treatment of RH ramies, resulting in decreased strength between the bonds and the DP of cellulose contained in the ramies, causing the ramies to break easily [44].

In addition to tensile strength, the evaluation of mechanical properties can also be performed by determining the value of the elastic modulus (MOE). Figure 15 and Table 3 show the average MOE results for both types of ramie, RN and RH, impregnated using tannin-based Bio-PU and tannin-based Bio-NIPU, respectively. It can be seen that RH had a higher MOE (9.1 GPa) than RN (6.7 GPa); this occurred due to the pre-treatment performed. After pre-treatment, the ramie became softer and had a higher MOE value due to the sap removal process. Figure 15 also shows a graph illustrating that impregnation using the tannin-based Bio-NIPU resin reduced the MOE values of both RN (11.7 GPa) and RH (6.2 GPa), compared to the use of the tannin-based Bio-PU resin on RN (13.5 GPa) and RH (11.7 GPa) (Table 3). These results are inversely proportional to the tensile strength (Figure 14); the inverse effects arise because the resin application process is used for non-elastic ramie materials, and thermosetting resins also cause the MOE of the ramies to decrease. However, it is still higher than for the non-impregnated ramies.

## 4. Conclusions

This study extracted tannin from mangium bark, yielding 26.43%. Then, the extracted tannins were used to prepare eco-friendly tannin-based Bio-PU and tannin-based Bio-NIPU resins, both used as impregnants for the modification of ramie fiber. Thermal and mechanical analysis showed that the tannin-Bio-NIPU and tannin-Bio-PU resins improved the ramie fiber’s thermal stability from around a residue of 7.3–18.9% to a residue of 19.1–30.5%, as well as the mechanical characteristics, including tensile strength (from around 200 MPa to be 451.3 MPa) and MOE (from 6.7 and 9.1 GPa to 13.5 GPa). The results demonstrated that tannin-based Bio-NIPU resin is a more effective impregnate than tannin-based Bio-PU resin as the former led to a higher weight gain in ramie fiber, owing to its lower viscosity and cohesion strength (respectively, 20.35 mPa·s and 5.08 Pa). Ramie impregnated with tannin-based Bio-NIPU resin exhibited higher tensile strength (approx. 426.1 to 451.3 MPa). Contrarily, ramie fiber impregnated with tannin-based Bio-PU resin displayed a better elastic modulus (11.7 to 13.5 GPa). Apart from this, the pre-treatment of ramie fiber with H_2_O_2_ also significantly influenced the properties of the impregnated fiber. Compared to natural ramie fiber, the pre-treated ramie fiber displayed lower tensile strength and elastic modulus after impregnation treatment. Both the tannin-Bio-NIPU and Bio-PU resins can improve the properties of ramie fiber; due to their eco-friendly characteristics, their industrial potential as a functional material with added value is promising. For further research, it is necessary to develop Bio-NIPU tannin-based resin formulations to provide the same or better thermal stability and mechanical strength results than Bio-PU tannin-based resins. In addition, it is also necessary to develop optimal pre-treatment formulations in order to obtain softer ramie fibers with good mechanical properties.

## Figures and Tables

**Figure 1 polymers-15-01492-f001:**
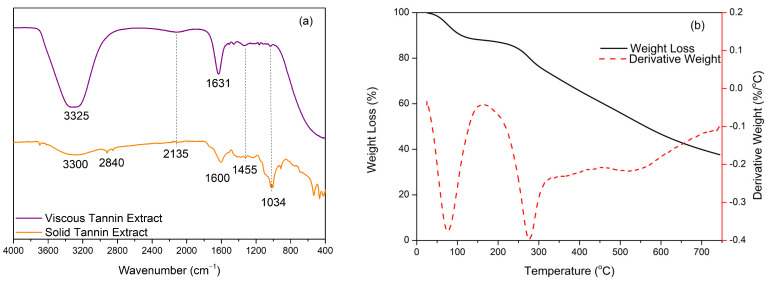
Properties of tannin extract: (**a**) FTIR (**b**) TGA/DTG.

**Figure 2 polymers-15-01492-f002:**
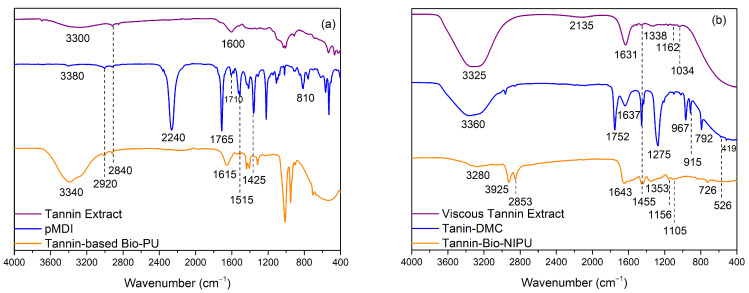
Spectra FTIR of tannin-based Bio-PU (**a**); tannin-based Bio-NIPU (**b**).

**Figure 3 polymers-15-01492-f003:**
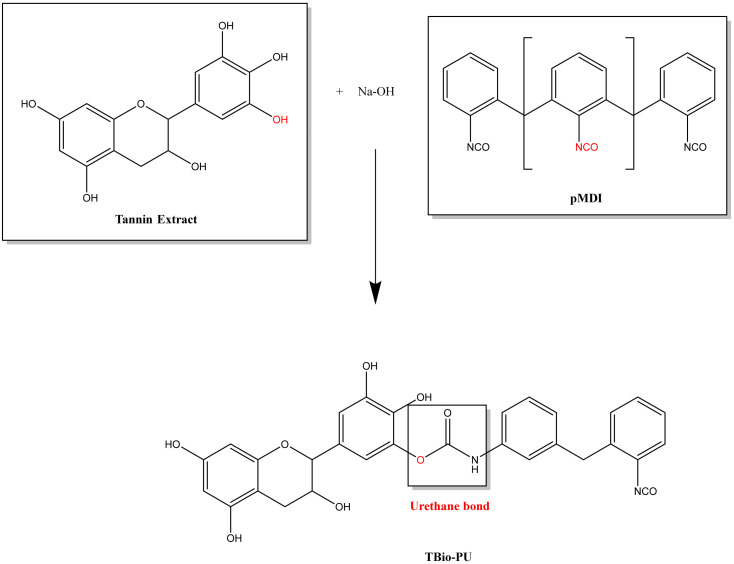
Possible reactions of tannin-based Bio-PU.

**Figure 4 polymers-15-01492-f004:**
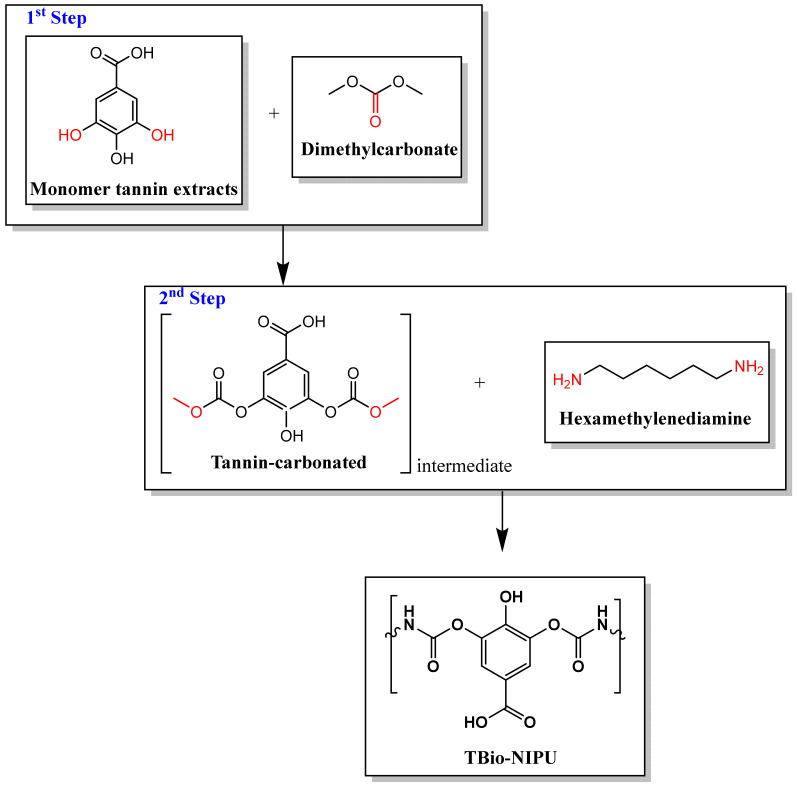
Possible reactions of tannin-based Bio-NIPU.

**Figure 5 polymers-15-01492-f005:**
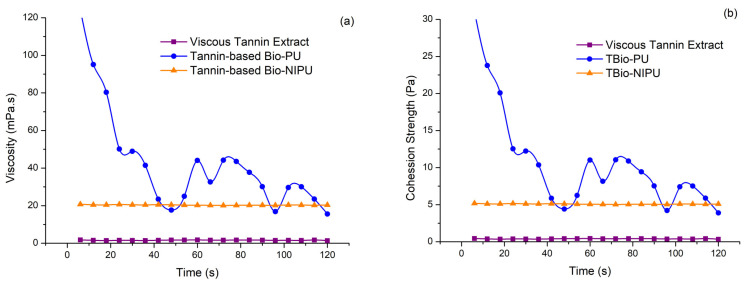
Physicochemical properties: (**a**) viscosity; (**b**) cohesion strength properties.

**Figure 6 polymers-15-01492-f006:**
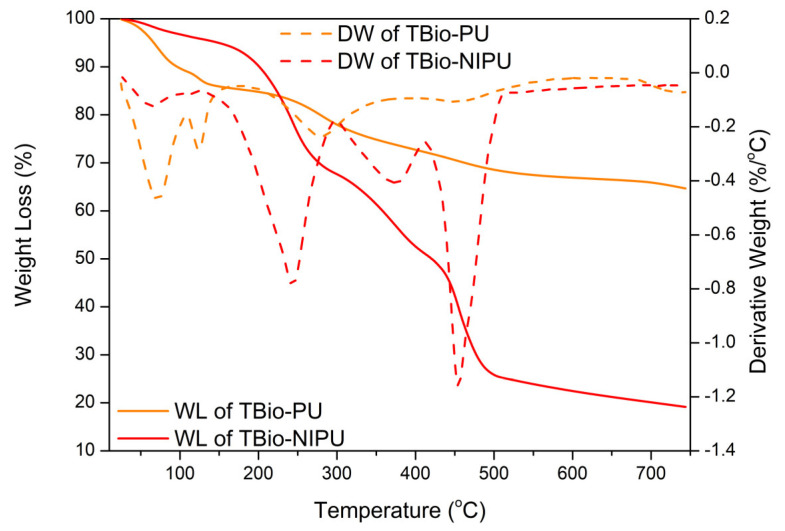
TGA/DTG curves of tannin-based Bio-PU and Bio-NIPU.

**Figure 7 polymers-15-01492-f007:**
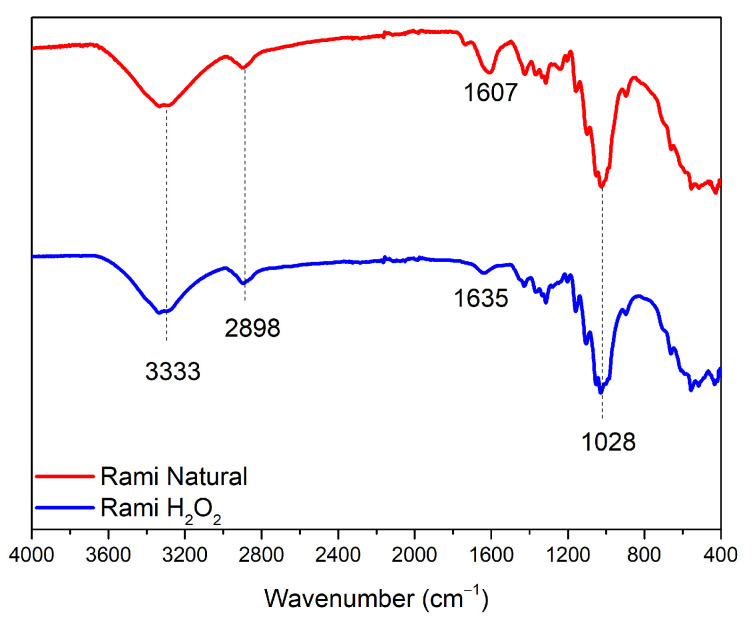
Typical FTIR spectra of different types of ramie fiber.

**Figure 8 polymers-15-01492-f008:**
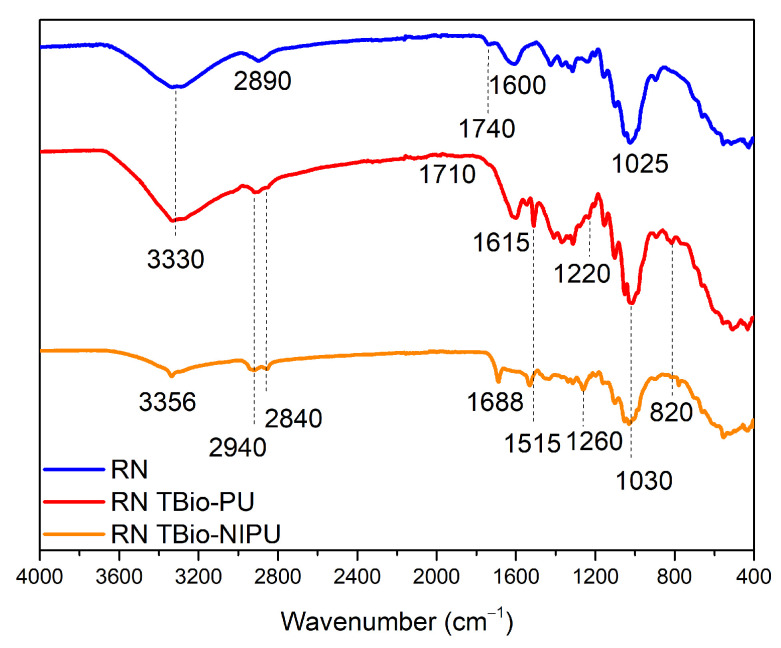
FTIR spectra of impregnated natural ramie.

**Figure 9 polymers-15-01492-f009:**
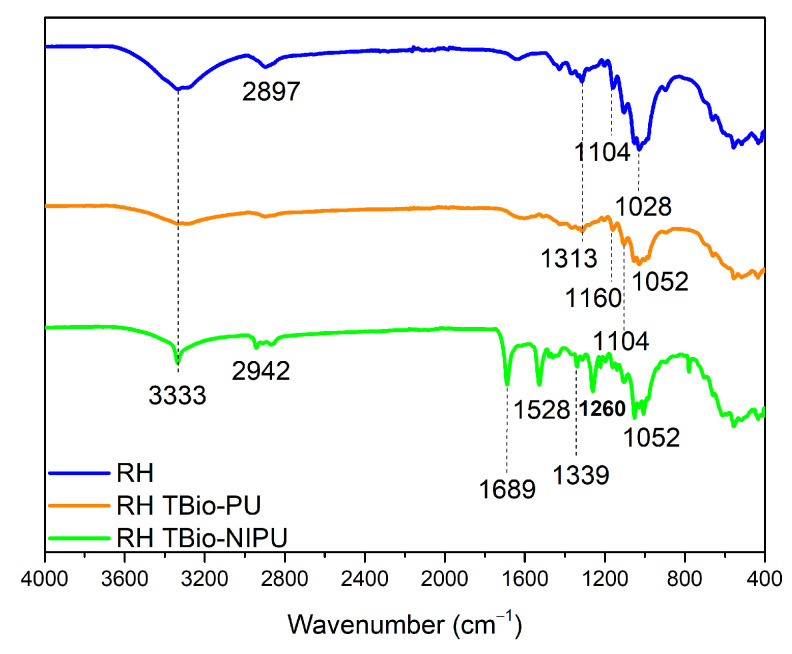
FTIR spectra of impregnated H_2_O_2_ pre-treated ramie fibers.

**Figure 10 polymers-15-01492-f010:**
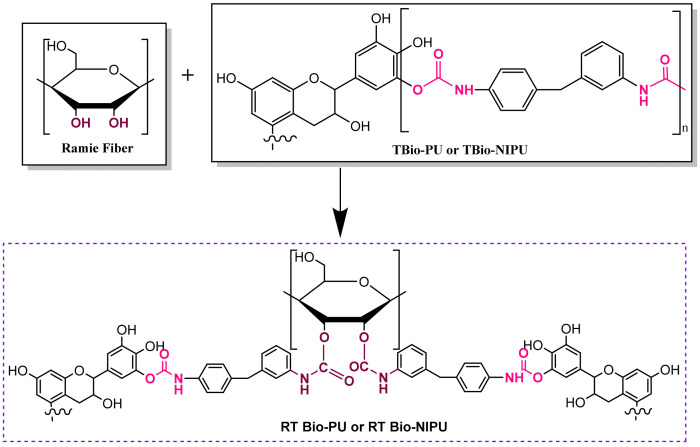
Possible mechanism reaction of ramie with resins.

**Figure 11 polymers-15-01492-f011:**
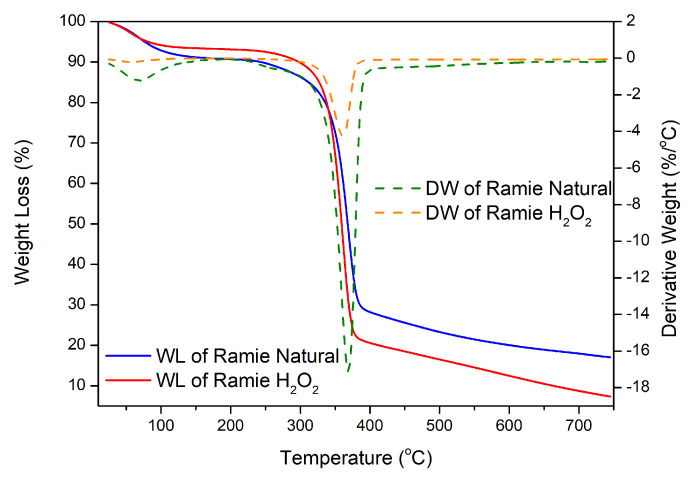
TGA/DTG curves of RN and RH.

**Figure 12 polymers-15-01492-f012:**
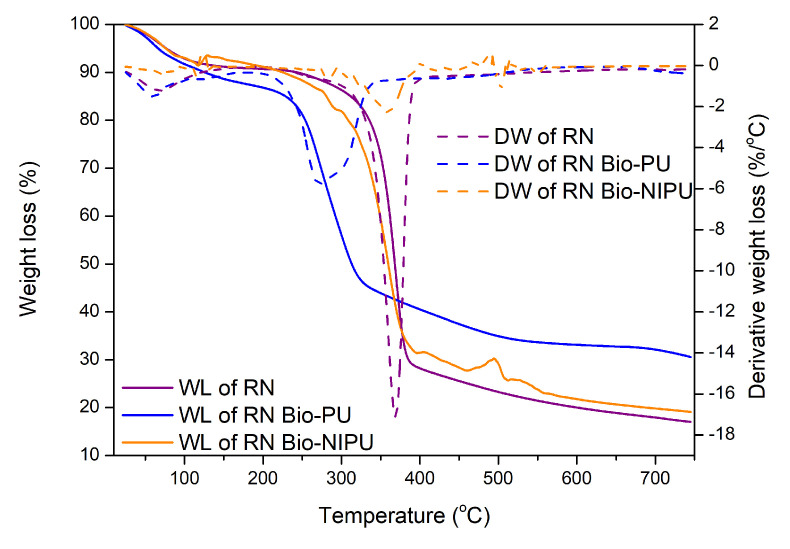
TGA/DTG curves of RN before and after impregnation.

**Figure 13 polymers-15-01492-f013:**
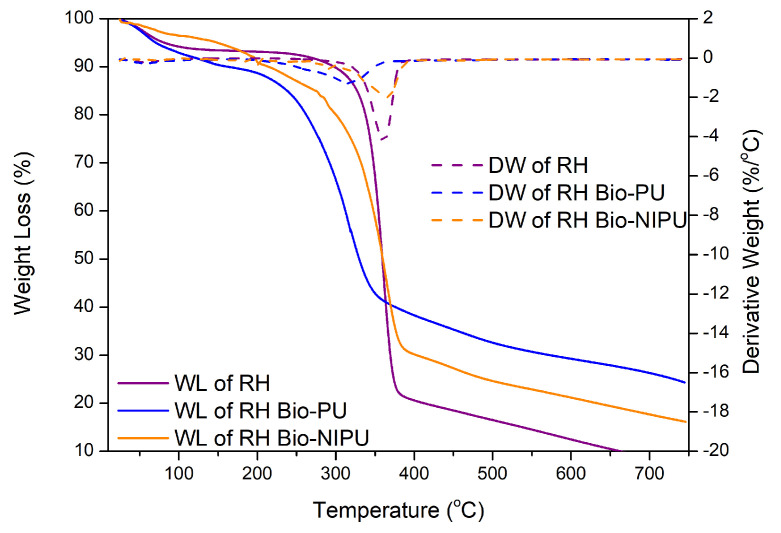
TGA/DTG curves of RH before and after impregnation.

**Figure 14 polymers-15-01492-f014:**
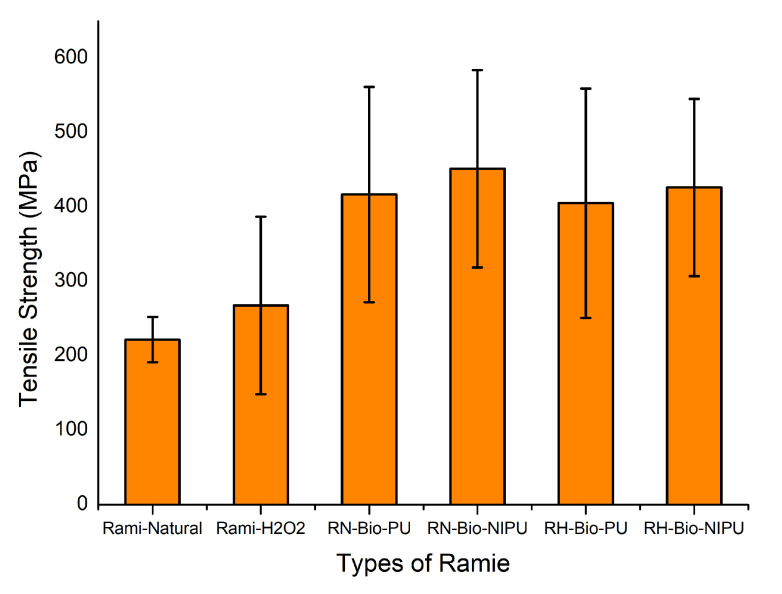
Tensile strength of different types of impregnated ramie with different resins.

**Figure 15 polymers-15-01492-f015:**
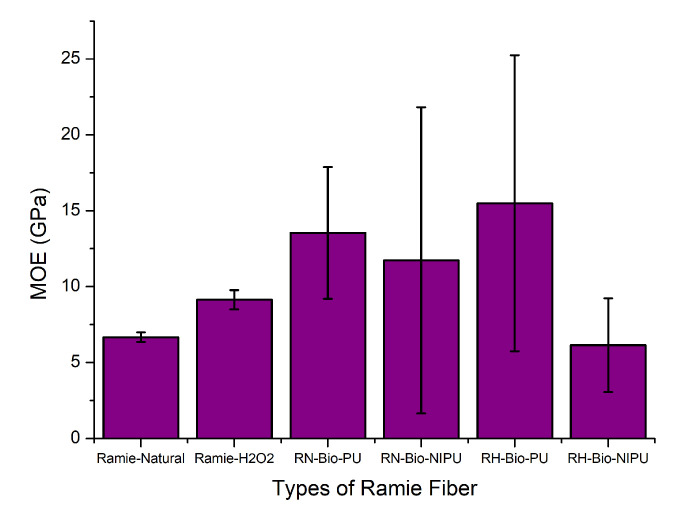
Modulus of elasticity (MOE) of different types impregnated ramie fibers.

**Table 1 polymers-15-01492-t001:** Characteristics of extracted tannins.

Parameter	Result	Reference
Moisture content of *Acacia mangium* (%)	5.47 ± 0.13	16.7 [20]
Yield of viscous tannin extracts (%)	26.43 ± 1.36	18.0 [25]
Solid content (%)	93.46 ± 0.41	89.8 [21]

**Table 2 polymers-15-01492-t002:** Weight gain of untreated and pre-treated ramie fiber impregnated with different resin types.

Type of Ramie Fiber	Resin Type	Weight Gain (%)
Ramie without pre-treatment (RN)	Tannin-based Bio-PU	18.9 ± 1.49
	Tannin-based Bio-NIPU	21.2 ± 3.58
Ramie with pre-treatment (RH)	Tannin-based Bio-PU	21.1 ± 4.73
	Tannin-based Bio-NIPU	26.5 ± 4.44

**Table 3 polymers-15-01492-t003:** Summarized results of tensile strength and MOE of ramie fibers.

Ramie Type	Resin Type	Tensile Strength (MPa)	MOE (GPa)
Ramie without pre-treatment (RN)	RN	221.63 ± 30.43	6.68 ± 0.31
	RN-Bio-PU	416.69 ± 144.71	13.54 ± 4.34
	RN-Bio-NIPU	451.25 ± 132.55	11.74 ± 10.07
Ramie with pre-treatment (RH)	RH	267.57 ± 119.17	9.14 ± 0.63
	RH-Bio-PU	404.94 ± 154.2	15.49 ± 9.75
	RH-Bio-NIPU	426.13 ± 119.2	6.5 ± 3.08

## Data Availability

The data presented in this study are available on request from the corresponding author.

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
