# Peer review of "Eco-Friendly Tannin-Based Non-Isocyanate Polyurethane Resins for the Modification of Ramie (Boehmeria nivea L.) Fibers"

_polymers, 2023, doi:10.3390/polym15061492_

Round 1

Reviewer 1 Report

Manuscript Title: Eco-Friendly Tannin-based Non-Isocyanate Polyurethane Resins for the Modification of Ramie (Boehmeria nivea L.) Fibers

The paper develosp a tannin-based non-isocyanate polyurethane (tannin-Bio-NIPU) and tannin-based polyurethane (tannin-Bio-PU) resins for impregnating ramie fibers (Boehmeria nivea L.) and investigate their mechanical and thermal properties.

The subject is interesting and the paper is well written and taught. Some minor revisions are needed before final acceptance

1. Some important findings should be clearly mentioned in the abstract

2. The strength of the current work should be clearly highlighted in the introduction part

3. Hydrogen peroxide (7%)? Are the authors sure that they used Hydrogen peroxide at this mentioned concentration?

4.  The sections Examination of Ramie Fiber Properties and Characterization of Viscous Tannin Extracts, Tannin-based Bio-PU, and Tannin-based Bio- 197 NIPU Resins should be merged into single paragraph

5. In Figure 6, the title should be TGA/DTG curves instead of thermal stability. Please do this for all TGA curves

6. Figure 7, the title is not suitable, please change

7. Table 2, please define the abbreviations RN and RH

8. The results given in Figures 14 and 15 should be given in a Table summarizing the results

9. In conclusion, please address a future recommendation

Reviewer 2 Report

The article analysis novel approach of non-isocyanate PUs. The article is interesting, well written and the idea is really novel. However, I have few remarks:

1. Abstract part misses the most important results expressed by numerical values.

2. In Introduction section, authors mention several studies regarding non-isocyanate-based PUs. Therefore, the question arises why authors suggested method is better compared to the ones which were already suggested. Can you please indicate the disadvantages of other methods if there are any and advantages your authors suggested method?

3.  Article contains Results section, but there is no Discussion section. I would like to suggest "Results and Discussion" section instead of only "Results" section.

4. Figure 14. How significant is the difference between RN-Bio-PU - RH-Bio-NIPU? I mean the scattering of the individual values is very broad and it overlaps the results of other PU NIPU compositions.

5. Conclusion section is also missing the main results expressed by numerical values. Where such synthesized PU can be applied industrially?

4. 

Round 2

Reviewer 2 Report

Authors have taken into consideration all my remarks.